# The Research of AHP-Based Credit Rating System on a Blockchain Application

Chao Chen [1,2,*,†], Hao Huang [1,†] , Bin Zhao [1], Desheng Shu [1] and Yu Wang [1]

1 College of Computer Science and Engineering, Sichuan University of Science and Engineering, Zigong 643033, China
2 Sichuan Key Provincial Research Base of Intelligent Tourism, Zigong 643033, China
* Correspondence: chenchao@suse.edu.cn
† These authors contributed equally to this work.

**Abstract:** NFT is a kind of virtual token derived from the blockchain. In 2019, the NFT transaction market became a new force in the field of the digital economy, while NFT fraud was also widespread. There is no efficient technology or methods to ensure the authenticity of the source data (which have not been stored on the blockchain yet) on a blockchain traceability system. To solve this problem and to safeguard the rights and interests of members of the blockchain application, we propose a method to measure the user's credit degree by obtaining the data before it stores on the blockchain. We first analyze some NFT trading markets' business processes and dealing models. Then, based on the analytic hierarchy process (AHP) in the operational research theory, some indexes of credit rating have been made. A credit rating system has been established by calculating the evaluation matrix and efficacy coefficient of each index. The experimental results show that the credit evaluation system can be used as a method to judge the user's credit rating on a blockchain traceability system. This method provides a reference for the decision of whether to restrict the transaction of some users with abnormal behavior.

**Keywords:** blockchain; non-fungible-token; analytic hierarchy process; credit rating; internet of things

## 1. Introduction

Blockchain is a data structure that is used to store data, and it has the characteristics of data decentralized storage, not being easy to tamper, and traceability [1,2]. These characteristics make blockchain technology not only limited to the financial field, but there are also many studies and applications in the internet of things industry, smart cities, logistics, justice, and other fields. In 2017, Ant Financial, a subsidiary of Alibaba (Hangzhou, China) Network Technology Co., Ltd., launched a blockchain traceability application [3], which is the first well known enterprise in China to penetrate blockchain application research and development. In 2020, the scale of China's blockchain industry reached 5 billion yuan, and the scale of investment and financing increased year by year. Among them, the investment and financing of blockchain in the digital asset industry accounted for 14% of the total, and the application of logistics traceability in vertical industries accounted for 36.8% [4,5]. In 2021, the global blockchain technology market was estimated at $5.92B and is expected to grow at a CAGR of 85.9% from 2022 to 2030 [6].

The trading application of blockchain, with its consensus algorithm, no longer needs the traditional third-party credit deposit to guarantee the behavior of both parties. Additionally, for all parties in the chain, trading data cannot be tampered with. These ensure the safety and reliability of the data. To a certain extent, it also provides a new technology solution, which solves the tricky problems of traditional industry regulation. With the development of blockchain technology in the market, the concept of the non-fungible token (NFT), which is different from classic cryptocurrencies, such as Bitcoin [1] and Ethereum [7],

has been created. As of 14 December 2022, the outstanding value of the NFT market has exceeded $40 billion [8].

### 1.1. The Introduction of NFT

NFT is a cryptocurrency derived from Ethereum's smart contracts that cannot be copied, replaced, or subdivided. NFT is recorded in the blockchain to prove the authenticity and ownership of digital products [9]. It is different from Bitcoin and other classic cryptocurrencies in internal characteristics [10]. Bitcoin is a divisible, homogeneous cryptocurrency with equal value. In contrast, NFT is a non-divisible, non-fungible token. Because of these features, NFT can be used for unique ID identification. Specifically, by using NFT, people can easily prove their ownership of photos, music, videos, domains [11], notes [12], financial derivatives [13], and even some physical assets, such as art, real estate, stamps, gold, etc.

### 1.2. The NFT Market Is Booming

Although the NFT is essentially a bunch of code, the buyer will think that the code is valuable and pay for the relative rarity of the digital asset, as well as its verifiable and trust-absent-approved transfer characteristics [9]. On OpenSea [14], 5841 ETH worth of transactions is made on an average day, compared to 9075 ETH on Blur [15,16]. In the year to 12 December 2022, there were more than 110 million NFT transactions, with a total value of more than $140B [17]. In addition, various NFT advisory websites (e.g., NonFungible [18], NFT Bank [19], DappRadar [8], Defi Pulse [20]) and a series of trading markets (e.g., OpenSea, Magic Eden [21], AtomicMarket [22]) also provide a relatively secure environment and sufficient transaction information. NFTMs make profits by charging gas fees for transaction fees. For example, the transaction fees generated by using ETH on the OpenSea platform account for more than 20% of the transaction fees in the whole Ethereum trading market [23]. In addition, the price trend of virtual currencies will also affect the sales volume of NFTs. By analyzing the price trends of bitcoin and Ethereum and the transaction volume of the NFT markets from 2018 to 2021, it was found that the price rising of Bitcoin and Ethereum will lead to an increase in NFT sales volume. The result shows that the cryptocurrency market will affect the growth and development of the NFT markets, but there is no reverse effect [24].

### 1.3. Security Issues of NFT Transactions

As the NFT market grows rapidly with millions of dollars in sales, criminals and scammers inevitably flood the market to defraud buyers and steal their digital assets. News of NFTs theft has also been common, and as of July 2021, more than $100 million of NFTs were stolen [25]. Some scammers use the unique features of smart contracts to create a Trojan horse NFT trap token that, if accepted, can immediately steal buyers' accounts [26]. In addition, NFT exchange scams often occur, in which scammers are tricked into accepting what looks like a "like-for-like" exchange by simply creating a digital asset with the same name and image as other high-value NFTs. The report NFTs and Financial Crime [27], released by Elliptic, describes seven types of NFT theft (see Table 1), which states that over 4600 NFTs were stolen on 14 July 2022 alone, which was the highest ever recorded. Legitimacy and security have always been a big problem for NFTs, as there is currently no good way to stop impostors from selling other people's art that the authors are all unaware of the fraud. So the situation is that people who create or list the NFT on the blockchain do not own the rights to the relevant digital assets, and if the real owners are unaware of the transactions, then the thief will benefit from the fruits of those owners' labor. For example, in August 2021, an impostor posing as British graffiti artist Banksy sold an NFT for $336,000 through the online NFT market [28]. In addition, artist Aja Trier posted that her/his work has been stolen and sold 86,000 times and complained that there may be robots stealing her works and uploading them to the NFTMs [29]. However, even if some scammers do not steal NFTs, there are a series of behaviors that harm the interests of consumers. For

example, ulterior motives cheat consumers by giving their NFT the same name, using the same image address, or using similar pictures with other famous NFTs [23].

**Table 1.** 7 scams of NFT.

| Scam | Description |
| --- | --- |
| Phishing Scams | Through a fake pop-up |
| Trojan Horse | Send buyers malicious NFTs |
| Impersonation Scams | Pretending to be support staff of NFT marketplaces |
| Swap Scams | Swap NFTs with fake ones |
| Marketplace Invite Scams | Provide a fake game invitation code |
| The Stolen NFT Market | Theft of digital asset accounts |
| Laundering | Bundled with other risky centralized trades |

*1.4. Ways to Prevent the Theft of NFT*

Due to the lack of knowledge in the field of blockchain and NFT, these artists who suffer from NFT scams cannot find a proper way to protect their rights. They can only unite those artists who have been infringed on to make a collective appeal [30]. On OpenSea's side, OpenSea allows the creation of NFTs using "inert mint" that allows users to list NFTs for sale without writing them to the blockchain first, and sellers don't pay the gas fee until the NFT is sold, allowing scammers to list as many stolen items as possible. While other NFT markets also allow for "inert mint", OpenSea's popularity and its imperfect vetting system make it an ideal place for robots to lurk. In response to these infringements, earlier last year, DeviantArt launched Protect [30], an image recognition tool that can identify relevant artworks on the NFT markets and alerts their true authorship. Many artists have found that a large number of works have been stolen after using this tool. If these artists want NFTMs to remove and delete infringing works, they must submit a deletion request to the platform according to the Digital Millennium Copyright Act (DMCA) [31]. However, due to a large number of infringing works, it can take weeks for platforms to remove the offending titles. OpenSea even fails to respond to the removal requests of those art bloggers with few followers [32]. These artists had no choice but to amplify the problem by posting their removal requests in the form of advertisements on the Google Images website to attract the platform's attention [33]. OpenSea later announced that it would severely limit the free NFT listing policy, claiming that the free listing policy resulted in over 80% of NFTs being made through plagiarism, while many NFT minters were unhappy with this statement, which led to OpenSea lifting the previous restriction. In this context, SnifflesNFT [34] was created as an image recognition tool that, similar to Protect [35], is dedicated to identifying stolen works of art. NFTM Rarible, on the other hand, reduces plagiarism by implementing a manually moderated verification system that encourages sellers and creators to link their social media accounts and prevents unverified sellers' NFT from showing up in searches [36].

With the news of more and more digital asset theft, NFTMs such as OpenSea also began to formulate relevant policies to regulate the behavior of NFT casting, and it would automatically hide those NFTs marked as "suspicious" [37]. However, this also caused many consumers to question whether the NFT they purchased came from a clear source. Many users began to question those NFTs on the shelf and reported them constantly. As a result, some NFTs not obtained illegally were removed from the shelves, which hindered the development of the NFT market. Then, NFTMs can only refer the challenge to the police and give them full authority to deal with it [38]. In May 2022, OpenSea launched a tool to detect fake NFTs, which uses image recognition technology coupled with manual screening to screen out the stolen NFT and maintains the account verification process [39].

In addition, some scholars have proposed NFT certification by using the STRIDE [40] threat and risk assessment method. They believe that when users mint or sell NFT, a potential attacker may fail user authentication or illegally obtain NFT directly by stealing

private keys. They suggest adding NFT authentication to smart contracts and using a cold wallet to prevent private key disclosure [9]. We have read their article and found that this method tries to control the risk in the life cycle of smart contracts. We believe that it is an efficient method to protect private keys, but this method, by adding logic to smart contracts, will cause the complexity of smart contracts to some extent and reduce the efficiency of the entire NFT transaction. At present, the NFT trading system is limited by the underlying consensus algorithm of blockchain, and the degree of transaction concurrency is greatly reduced (Bitcoin reaches merely 7 TPS, while Ethereum only has 30 TPS [41]). In addition, it has been suggested that NFT projects should make the NFT open source with external tokens before it is minted on the platform to ensure that the NFT is not unknown or buggy, but currently, no NFT project supports this [23]. Existing solutions to NFT thefts are summarized in Table 2.

**Table 2.** Solutions to the NFT theft.

| No. | Author | Solution |
|-----|--------|----------|
| 1 | Artist | Appeal/DMCA/Advertise/Report to the police |
| 2 | DeviantArt | Image recognition: Protect |
| 3 | Snifflesnft.com | Image recognition: SnifflesNFT |
| 4 | Rarible | Manual verification |
| 5 | OpenSea | Mark and hide/Image recognition/Manual verification |
| 6 | Shostack, Adam | NFT validation/Cold wallet technology |
| 7 | Das, Dipanjan | External token open source review |

*1.5. Research in Other Blockchain Applications' Supervision*

In terms of the regulation of blockchain applications, Sun Linhui et al. proposed a diversified cooperative regulation mode for coal-mine safety based on blockchain technology [42]. By establishing a consortium blockchain with customs as the core, Mei Ao et al. realized the whole process supervision of cross-border e-commerce and improved the service capacity and supervision level [43]. Yong Binbin et al. proposed a reliable vaccine supply regulation system based on blockchain and used machine learning to provide people with more effective epidemic prevention methods [44]. Wang Yu et al. found abnormal nodes by detecting abnormal behaviors of user-control nodes in blockchain applications, and then based on users' information, associated the addresses of suspicious nodes in the transaction layer of the blockchain with the real addresses to realize the supervision of blockchain applications [45]. Xueping Liang et al. have designed and implemented an architectural ProvChain that collects and validates cloud data sources by embedding metadata into blockchain transactions, which can enhance privacy security and data availability [46]. Sidra Malik et al. proposed a blockchain traceability framework, ProductChain, which uses a three-layer fragmented architecture to ensure the availability of data to consumers, limit peers' query of information, and control the time of consumers' query of logistics information within milliseconds [47]. In order to prevent the circulation of counterfeit electronic hardware products in the market, Pinchen Cui et al. developed a supply chain traceability system named Commercial Off-the-Shelf (COTS) for micro-electronic components based on Hyperledger [48]. Uzair Javaid et al. proposed that, in the construction of a smart city, to ensure the consistency of data, they used physical unclonable functions (PUFs) and Ethereum, which could customize a smart contract, to establish data fingerprint and defend against data tamper [49]. Table 3 summarizes the methods for the supervision of blockchain applications described above.

**Table 3.** Supervision of blockchain applications.

| No. | Author | Method |
|-----|--------|--------|
| 1 | Mei Ao | Consortium Blockchain |
| 2 | Binbin Yong | Machine learning |
| 3 | Yu Wang | Detects abnormal behavior of control nodes |
| 4 | Xueping Liang | Collect and verify cloud data sources |
| 5 | Sidra Malik | Three-layer fragmented architecture |
| 6 | Pinchen Cui | Microelectronic component control |
| 7 | Uzair Javaid | PUFs |

To summarize the above papers' work, it can be found that for the protection of NFT, blockchain-related technology can only protect the data which is already stored on the blockchain, and the existing research is basically focused on the blockchain or the improvement of smart contracts. There are a few considerations for blocking the risk from every beginning. Therefore, to further protect the actual rights and interests of most of the parts in the application of blockchain transactions, this paper analyzes the NFT trading platforms, such as OpenSea, Magic Eden, and AtomicMarket from the perspective of controlling data anomalies at the initial end of the blockchain, and then summarizes the hierarchical structure trading models of the above NFT trading platforms. Most importantly, a user credit rating evaluation system is established by using the analytic hierarchy process (AHP) in operations research. The credibility of users on the chain can be judged by using the credit rating evaluation system. Additionally, the result of the credit assessment can also be used as the standard to regulate and restrict users' other operations, which need higher credit on the blockchain.

*1.6. Introduction to Analytic Hierarchy Process*

As a multi-standard decision-making tool, the analytic hierarchy process (AHP) has been applied in all decision-making-related studies (sociology, anthropology, education, manufacturing, politics, engineering industry, etc.) [50]. It decomposes elements related to decision-making into levels, such as objectives, criteria, and schemes, which are simple, flexible, and practical [51].

*1.7. Research Purpose and Structure of This Article*

In view of the phenomenon of NFT theft, we explore why there are so frequent NFT theft phenomena from the view of a bystander of these NFTMs. In the past, many scholars look forward the solutions from blockchain consensus or smart contracts, while we consider from the outside of the blockchain system and hope to adopt a way to prevent the behavior of stealing NFT at the very beginning of the transaction process. Through extensive research, we are the first team to combine the field knowledge of operations research with the NFT industry. In our opinion, to prevent the purpose of NFT theft, we calculate the user's credit rating according to the user's previous behaviors on the NFTMs and then decide whether to restrict some operations of the user (such as uploading NFT and selling NFT, etc.) according to the rating.

In Section 2, this article will analyze and establish the software hierarchical structure of the NFTM. In Section 3, the user credit rating evaluation model is established. In Section 4, our model is verified by experiments. In Section 5, we do a discussion about our experiments. Finally, it is summarized in Section 6.

**2. Analysis of NFT Trading Platform Model**

We used DappRadar, a popular dApp tracker, and selected some NFTMs that use Ethereum as their virtual currency. We mainly analyze the UAW (unique active wallet), transaction number, and transaction volume of these platforms in the past 30 days (see Table 4).

**Table 4.** DApp Average transaction data for 30 days.

| DApp | UAW/K | Transactions/M | Volume |
|---|---|---|---|
| OpenSea | 201.16 | 924.51 | $169.69 M |
| Magic Eden | 187.63 | 36.59 | $79.08 M |
| AtomicMarket | 89.79 | 8.59 | $2.42 M |
| Blur | 31.63 | 93.20 | $92.68 M |
| JPG Store | 29.16 | 7.59 | $8.41 M |
| X2Y2 | 13.98 | 34.00 | $55.95 M |

As OpenSea, Magic Eden, and AtomicMarket are the most popular NFT trading platforms in the world, the transaction volume of these platforms in the past 90 days occupied the first place, and each user had an average of 2.2, 5.5, and 1.8 transactions. Therefore, we mainly analyze the transaction information on these three platforms.

We analyzed the transaction events related to these platforms. On these platforms, each transaction includes the seller's address (current owner), the buyer's address (new owner), the price of the NFT which was sold, and the time that the owner transferred the NFT.

We mainly analyze OpenSea, Magic Eden, AtomicMarket, and other NFT trading platforms and summarize the hierarchical software structure of the above NFT trading platforms (see Figure 1).

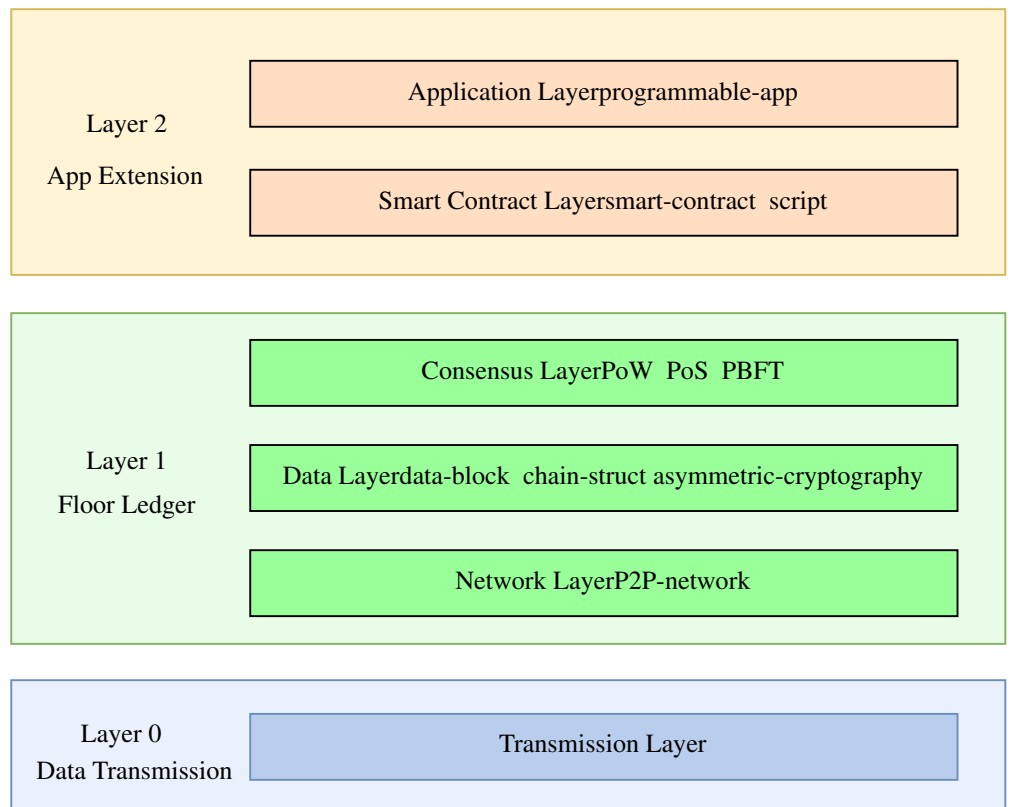

**Figure 1.** Hierarchical structure diagram of NFT trading platform.

In the app extension layer, the platform binds with the user's virtual currency wallet's account through the smart contract and script code. Additionally, it records the time and address of the user node on the blockchain when the user joins the platform. After users sign and activate these accounts, all the platform operations (NFT minting, transaction, transfer, etc.) will be tied to the account under the user's wallet and leave traces. When a user performs an NFT operation, the platform records the operator's address, NFT price, a

destination address, and date. This information can be used as the basis for the subsequent establishment of the user credit rating model.

## 3. The Establishment of a Credit Rating Evaluation Model

### 3.1. Analyze the Affecting Factors of Credit Rating and Establish the Hierarchical Structure Model

According to OpenSea and other digital art trading platform models and the above transaction information mentioned in Section 2, the elements that can be used to evaluate the user's credit rating are classified into the following three levels according to the analytic hierarchy process (AHP) standard (see Figure 2).

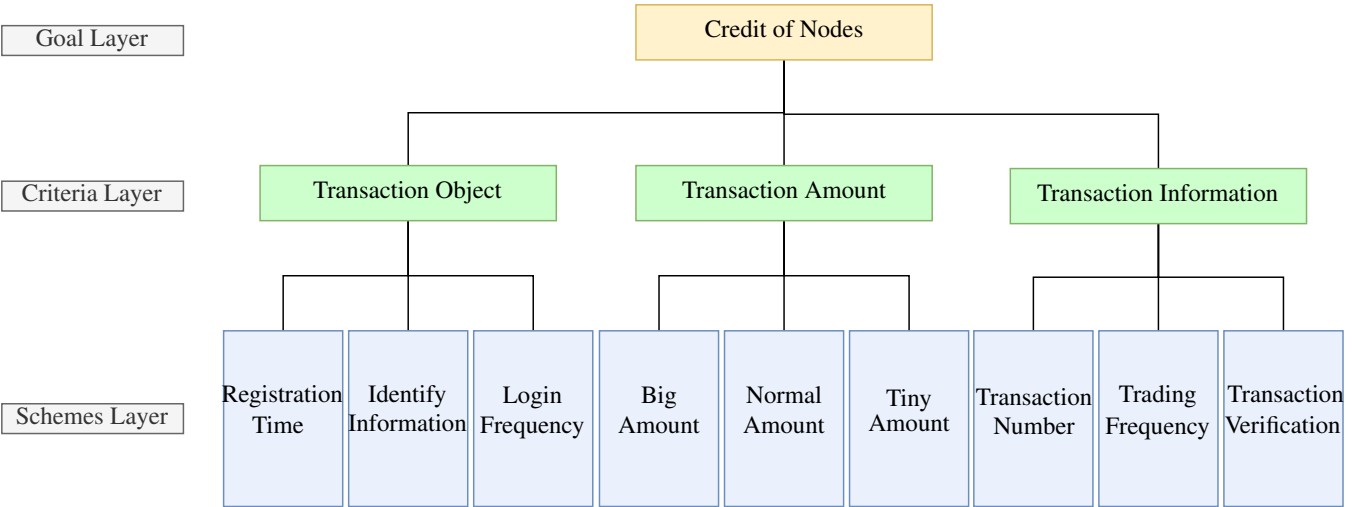

**Figure 2.** The three-tier structure of credit ratings.

The credit rating evaluation model is divided into three levels: goal layer (node's credit), criteria layer (transaction object, transaction amount, transaction information), and schemes layer (transaction object corresponds to registration time, identity information, login frequency; transaction amount corresponds to big amount transaction, medium amount transaction, small transaction; transaction information corresponds to transaction number, trading frequency, transaction verification). According to the hierarchical structure, there are three evaluation factors of the node credit rating evaluation model: transaction object, transaction amount, and transaction information. The node credit's evaluation matrix of the goal layer is $A = (a_{ij})_{(n \times n)}$, and the evaluation matrix of the transaction object, transaction amount, and transaction information in the criterion layer is $B_i$. Because the weights of each criterion in the criterion layer are different when measuring the credit rating, according to the definition of the scale of the evaluation matrix, the evaluation matrix of the credit rating evaluation of the system is as follows [51]:

$$A = \begin{pmatrix} a_{11} & a_{12} & a_{13} \\ a_{21} & a_{22} & a_{23} \\ a_{31} & a_{32} & a_{33} \end{pmatrix}, a_{ij} = 1/a_{ji}$$

Calculate consistency index ($CI$) according to the evaluation matrix:

$$CI = \frac{\lambda_{max} - n}{n - 1} \tag{1}$$

In the above Equation, $\lambda_{max}$ is the largest eigenvalue of the evaluation matrix. When the rank of a matrix is 3, the average random consistency index($RI$) of AHP is 0.52 [52].

Then, calculate the consistency ratio (*CR*) by the following formula:

$$CR = \frac{CI}{RI} \tag{2}$$

When $CR < 0.10$, it is considered that the consistency of the evaluation matrix is acceptable, otherwise the evaluation matrix should be modified.

### 3.2. Calculate the Weight Vector W

The calculation methods of weight vector include the geometric mean method, arithmetic mean method, eigenvector method, and the least squares method. Because each column in the evaluation matrix *A* approximately reflects the distribution of weights, therefore, we decide to calculate the average of all column vectors to estimate the weight vector. The specific formula is as follows:

$$W_i = \frac{1}{n}\Sigma_{j=1}^{n}\frac{a_{ij}}{\Sigma_{k=1}^{n}a_{kj}}, i = 1,2,...,n \tag{3}$$

The weight vector gives the relative importance of each index, and the larger the value, the more important the index, and the sum of the coefficient of every weight vector is 1.

### 3.3. Calculate the Total Random Consistency Ratio of Each Item in the Criterion Layer to the Goal Layer

Assuming that the relative weight of each index in the criterion layer is $b_i$, the following formula is used to calculate the total random consistency ratio of the scheme layer:

$$CR = \frac{\Sigma_{i=1}^{n}b_iCI_i}{\Sigma_{i=1}^{n}b_iRI_i} \tag{4}$$

When $CR < 0.10$, the random consistency requirement was met.

### 3.4. Determine Evaluation Vector $x_{ij}$ and Evaluation Matrix

The element of *i* represents the evaluation index of the criterion layer, element *j* represents the evaluation index of the scheme layer, and the system evaluation vector is defined as:

$$x_{ij} = V_n \cdot W_i \cdot w_j \tag{5}$$

We define that the evaluation model is divided into five credit levels:

$$V = \{excellent, prettygood, good, general, poor\} = \{V_1, V_2, V_3, V_4, V_5\}$$

And the scores are 100, 90, 75, 60, 40 respectively. The defined reference level is not a range level, but an indicator for calculating a user's credit membership. In Formula (5), $W_i$ is the weight of each index of the criterion layer to the goal layer, and $w_j$ is the weight of each index of the scheme layer to the criterion layer.

### 3.5. Calculate the Efficacy Coefficient of Each Credit Rating

The calculation formula is as follows:

$$d_i = [\Sigma_m^n(x_{mi} \cdot w_m)]W_i/2 \tag{6}$$

The total efficiency coefficient is:

$$D = \sqrt[n]{\prod_{1}^{n}d_i} \tag{7}$$

### 3.6. Calculate the Membership Degree of the Credit Ratings by the Total Efficacy Coefficient

Using the algorithm of Section 3.5, the standard total efficacy coefficient table of credit rating can be obtained. For different users, after getting the users' credit scores, the total efficacy coefficient of the users' credit scores can be calculated. For getting the users' final credit rating, it needs to calculate the membership degree among the standard total efficacy coefficient table by the users' total efficacy coefficient. Membership degrees can be judged in the following ways [53]:

We define that, for the rating of $i$, the membership degree can be calculated as follows:

$$\mu(y) = \frac{y - y_i}{y_{i+1} - y_i}, (y_i < y < y_{i+1}) \tag{8}$$

Additionally, for the rating of $i + 1$, the membership degree can be calculated as follows:

$$\mu'(y) = 1 - \mu(y) \tag{9}$$

If $\mu(y) < \mu'(y)$, users' credit ratings are near the rating of $i$ in the standard table, otherwise the users' credit ratings are near to the rating of $i + 1$.

## 4. Experiment and Result

According to the credit rating model in Section 3.1, the first-level indicators of credit evaluation are formulated as follows: transaction object $U_1$, transaction amount $U_2$ and transaction information $U_3$; the secondary indicators are registration time $U_{11}$, identity information $U_{12}$, login frequency $U_{13}$, big amount $U_{21}$, normal amount $U_{22}$, tiny amount $U_{23}$, transaction number $U_{31}$, trading frequency $U_{32}$, and transaction verification $U_{33}$. The users' credit rating rules are defined as follows (see Table 5).

**Table 5.** Rules of credit rating.

| First-Level Indicators | Second-Level Indicators | Evaluation | Details |
|---|---|---|---|
| | $U_{11}$ | Registration time | The score increases by 1 as the year increases |
| $U_1$ | $U_{12}$ | Identity information: consumer, minter & both | Consumer gets 2 score; minter gets 10 score; both get 15 score |
| | $U_{13}$ | Login frequency | Login per year: 1 score; login per month: 5 score; login per week: 10 score |
| | $U_{21}$ | Big amount(In the case of the ETH) | ETH[10,+∞): 20 score |
| $U_2$ | $U_{22}$ | Normal amount | ETH[1,10): 5 score |
| | $U_{23}$ | Tiny amount | ETH[0,1): 2 score |
| | $U_{31}$ | Transaction number | [100,+∞): 20 score; [10,100): 10 score; [0,10): 2 score |
| $U_3$ | $U_{32}$ | Transaction frequency | Deal per week: 20 score; deal per month: 10 score; deal per year: 2 score |
| | $U_{33}$ | Transaction validation | Platform has bound with wallet account: 3 score; nor: 0 score |

The evaluation matrix ($A$) of the first-level index is given as follows:

$$A = \begin{pmatrix} 1 & 1/3 & 1/5 \\ 3 & 1 & 1/2 \\ 5 & 2 & 1 \end{pmatrix}$$

Calculate the maximum eigenvalue of the evaluation matrix by MATLAB: $\lambda_{max} = 3.0037$. The algorithm (see Algorithm 1) to calculate the maximum eigenvalues of the matrix is:

---

**Algorithm 1:** Calculate for the max eigenvalue of matrix.

1 function [m,n,z] = fun(x)
2     [m,n] = eig(x);
3     diagN = diag(n);
4     z = max(diagN);
5 end

---

In the above algorithm, $n$ is the diagonal matrix formed by the eigenvalues of matrix $x$, $m$ is the eigenvector corresponding to the eigenvalues, and $z$ is the largest eigenvalue of the matrix.

The consistency index and consistency ratio are calculated as follows: $CI = 0.0019$, $CR = 0.0037$. Since $CR$ is less than 0.10, the evaluation matrix meets the calculation conditions.

The weight vector of the evaluation matrix calculated by the arithmetic mean method is:

$$A = (0.1095, 0.3092, 0.5813)^T$$

The obtained vector coefficients represent the relative importance of the first-level indicators.

Similarly, the evaluation matrix of each second-level index, whose CR is less than 0.10 is set as transaction object ($B_1$), transaction amount ($B_2$), and transaction information ($B_3$):

$$B_1 = \begin{pmatrix} 1 & 1/2 & 1/5 \\ 2 & 1 & 1/5 \\ 5 & 5 & 1 \end{pmatrix}$$

$$B_2 = \begin{pmatrix} 1 & 2 & 3 \\ 1/2 & 1 & 2 \\ 1/3 & 1/2 & 1 \end{pmatrix}$$

$$B_3 = \begin{pmatrix} 1 & 1/3 & 1/7 \\ 3 & 1 & 1/5 \\ 7 & 5 & 1 \end{pmatrix}$$

The maximum eigenvalues of each evaluation matrix are 3.0536, 3.0092, and 3.0649 respectively. $CIs$ are 0.0268, 0.0046 and 0.0325, respectively. $CRs$ are 0.0479, 0.0082 and 0.0625, respectively.

The weight vector of each evaluation matrix is:

$$B_1 = (0.1150, 0.1822, 0.7028)^T$$

$$B_2 = (0.5390, 0.2973, 0.1637)^T$$

$$B_3 = (0.0833, 0.1932, 0.7235)^T$$

According to the weight vectors of the goal layer and the criterion layer, the evaluation matrix (see Table 6) is determined by the Formula (5) in Section 3.4.

Calculate the efficiency coefficient (take $V_1$ as an example): $d_{B_1} = 0.1904, d_{B_2} = 1.2490, d_{B_3} = 6.3643$.

Calculate the total efficiency coefficient (take $V_1$ as an example): $D_{V_1} = 1.1481$.

Similarly, the total efficiency coefficient of the remaining credit ratings (see Table 7) can be obtained as: $D_{V_2} = 1.0036, D_{V_3} = 0.8379, D_{V_4} = 0.6870, D_{V_5} = 0.4596$.

**Table 6.** The evaluation matrix of credit rating.

|  | $V_1$ | $V_2$ | $V_3$ | $V_4$ | $V_5$ |
|---|---|---|---|---|---|
| $C_1$ | 1 | 1 | 1 | 1 | 1 |
| $C_2$ | 1 | 1 | 1 | 1 | 1 |
| $C_3$ | 7 | 6 | 5 | 4 | 3 |
| $C_4$ | 16 | 14 | 12 | 10 | 6 |
| $C_5$ | 9 | 8 | 6 | 5 | 3 |
| $C_6$ | 5 | 4 | 3 | 3 | 2 |
| $C_7$ | 4 | 4 | 3 | 2 | 1 |
| $C_8$ | 11 | 10 | 8 | 6 | 4 |
| $C_9$ | 42 | 37 | 31 | 25 | 16 |

To summarize, the efficacy coefficient of the credit rating is as follows.

**Table 7.** Semantic description and Efficiency Coefficient of credit rating.

| Rating | Semantic Description | Efficiency Coefficient |
|---|---|---|
| $V_1$ | Excellent | 1.1481 |
| $V_2$ | Pretty good | 1.0036 |
| $V_3$ | Good | 0.8379 |
| $V_4$ | General | 0.6870 |
| $V_5$ | Poor | 0.4596 |

## 5. Discussion

We collected the transaction data of more than 100 users on the OpenSea platform as the data set for our model experiment. After calculating and analyzing these data, it is found that this model can accurately reflect the credit degree of these users.

Next, the first 20 pieces of data in the data set are taken for verification (see Table 8). According to the scoring criteria in Table 5 and the method of calculating the efficacy coefficient in Section 3, the credit ratings of the goal layer indicator of the three users are calculated as.

**Table 8.** User Efficacy Coefficient of credit rating, Total Efficacy Coefficient, and credit rating.

| User | Score | $d_{B_1}$ | $d_{B_2}$ | $d_{B_3}$ | $D$ | Rating |
|---|---|---|---|---|---|---|
| 1 | 45 | 0.1458 | 0.8741 | 4.3163 | 0.8194 | Good |
| 2 | 53 | 0.1717 | 1.0279 | 5.0837 | 0.9645 | Pretty good |
| 3 | 14 | 0.04535 | 0.2720 | 1.3429 | 0.2549 | Poor |
| 4 | 13 | 0.04211 | 0.2521 | 1.2469 | 0.1151 | Poor |
| 5 | 38 | 0.1231 | 0.7370 | 3.6449 | 0.5750 | General |
| 6 | 25 | 0.08099 | 0.4848 | 2.3980 | 0.3069 | Poor |
| 7 | 36 | 0.1166 | 0.6982 | 3.4531 | 0.5302 | Poor |
| 8 | 26 | 0.08423 | 0.5042 | 2.4939 | 0.3254 | Poor |
| 9 | 13 | 0.04211 | 0.2521 | 1.2469 | 0.1151 | Poor |
| 10 | 42 | 0.1361 | 0.8145 | 4.0286 | 0.6682 | General |
| 11 | 48 | 0.1555 | 0.9309 | 4.6041 | 0.8164 | Good |
| 12 | 40 | 0.1296 | 0.7757 | 3.8367 | 0.6210 | General |
| 13 | 30 | 0.09718 | 0.5818 | 2.8776 | 0.4034 | Poor |
| 14 | 97 | 0.3142 | 1.8812 | 9.3041 | 2.3452 | Excellent |
| 15 | 23 | 0.07451 | 0.4461 | 2.2061 | 0.2708 | Poor |
| 16 | 31 | 0.1004 | 0.6012 | 2.9735 | 0.4237 | Poor |
| 17 | 43 | 0.1393 | 0.8339 | 4.1245 | 0.6922 | General |
| 18 | 22 | 0.07127 | 0.4267 | 2.1102 | 0.2533 | Poor |
| 19 | 39 | 0.1263 | 0.7564 | 3.7408 | 0.5979 | General |
| 20 | 22 | 0.07127 | 0.4267 | 2.1102 | 0.2533 | Poor |

Specifically, User 3 has been registered on the OpenSea platform for more than two years. As a NFT consumer on the OpenSEA platform only, he logged in every year and made only four transactions in the past two years, and the total transaction volume was less than 1 ETH. After the calculation of the credit rating model algorithm with the above method, the user's credit rating result is poor.

After analyzing the data set, it can be concluded that 53.77% of the around 100 users in the data set have a credit rating of V4 or lower, 18.87% of those who have a credit rating of V3 or above, and only 3.77% users have a credit rating of V1. Based on this data, the platform can develop a more detailed user rights scheme.

It can be seen that the model can integrate and analyze the relevant historical data of users to a certain extent, and the results can be used as the criteria for restricting users to do operations of higher credit levels in the future.

## 6. Conclusions

Among the blockchain applications in China, financial industry applications account for 41% of the total applications [5]. To a certain extent, the data tamper-proof property of blockchain technology itself ensures the security of the data that already be stored on the blockchain. However, there is no good method to judge and supervise the authenticity of metadata before it is stored on the chain [54]. We use the analytic hierarchy process (AHP) to rate the credit of users on the blockchain trading platform. The result can be used as a benchmark to determine whether users can subsequently perform certain operations with higher credit levels required on the blockchain. It verifies that the evaluation system established by the method proposed in our work has a certain guiding significance. However, there is a disadvantage to this method, which is deeply bound to blockchain applications. For different applications, there will be different credit rating scoring methods, so of course, the evaluation matrix will be naturally different. For the regulation of data security and users' credit on blockchain trading platforms, whether to continue to adopt the traditional third-party supervision and posterior punishment or to establish a set of industry standards for trusted operation on the consortium blockchain is a further research direction.

Aiming at the problem of NFT scams, we put forward the idea of blocking scam behavior at the forefront of the transaction process and used the AHP method of operations research to conduct modeling and experiments. Through analyzing the model data, it can be seen that this model can conduct credit ratings according to users' past transactions and other behaviors on the platforms, which provides a reference for whether to restrict users' trading operations in the future. However, there are still some shortcomings in our research. Through data analysis, we find that the standards for user credit rating are not detailed enough to fully match the NFT transaction business. In addition, the credit evaluation system designed in this paper is so strict for new users that the initial credit rating of new users will be very low, which may not be conducive to the platform to attract new users to join. These questions need further study in the future.

**Author Contributions:** Conceptualization, H.H.; methodology, H.H.; software, H.H.; validation, H.H., C.C. and B.Z.; formal analysis, C.C. and B.Z.; investigation, H.H. and D.S.; resources, H.H.; data curation, H.H. and C.C.; writing—original draft preparation, H.H.; writing—review and editing, H.H. and Y.W.; visualization, H.H. and Y.W.; supervision, C.C.; project administration, H.H. and C.C.; funding acquisition, C.C. All authors have read and agreed to the published version of the manuscript.

**Funding:** This work was supported by the National Natural Science Foundation of China (No. 42074218) and the Sichuan Key Provincial Research Base of Intelligent Tourism (No. ZHZJ22-02).

**Institutional Review Board Statement:** Not applicable.

**Informed Consent Statement:** Not applicable.

**Data Availability Statement:** Not applicable.

**Conflicts of Interest:** The authors declare no conflict of interest.

**Abbreviations**

The following abbreviations are used in this manuscript:

| | |
|---|---|
| NFT | Non-Fungible-Token |
| NFTM | NFT Market |
| AHP | Analytic Hierarchy Process |
| CI | Consistency Index |
| CR | Consistency Ratio |

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
