# Peer review of "The Research of AHP-Based Credit Rating System on a Blockchain Application"

_electronics, doi:10.3390/electronics12040887_

Round 1
Reviewer 1 Report
The review of sources in the 2nd paragraph of Introduction describes only references of authors from China. There are huge of methods and projects for creating reliable NFTs based on platforms such as Ethereum, Solana, Cardana ... in the 20s
The authors proposed a method for the users' credit rating evaluation. Proposed parameters and the calculation of the overall assessment looks argumentative.
The description of the experiment, in which only 3 users are listed, is confusing - is one of them a real scammer? Considering such a small data sample is impossible to draw any conclusions.
Another serious flaw:
- The size of the article is only 8 pages. Minimum 14 pages needed.
- Dot at the end of headings
- There are only 13 sources at the end and most of them are not formatted - it is impossible to find these articles and analyze them
Reviewer 2 Report
1. The first line of the abstract need to correct. Also, none of the sentences will start with "And" therefore revise it in the abstract. Also mention results and improvement in the abstract. The introduction section needs to be improved, adding some basic detail and pictorial representation should be presented in this section. Add subsection of contribution and organization
2. The concept of paper is good. The vast literature study in regard to methodology and techniques is documented in the manuscript. Still a table of a complete survey of the last decade would be suggested.
3. Highlight the improvement of the proposed technique in form of percentage improvement over the existing techniques.
4. Separate the result and discussion and detail every figure and table properly. The quality of the tables and figures should be improved. Increase the DPI of all the figures (A few larger words are requested in figures).
5. Results analyses are good and very well presented in terms of tables and graphs. Discussion on obtained results is effective. Comparing the presented technique with the state-of-the-art techniques would show the effectiveness of the proposed work which shows the novelty. It is requested to highlight it.
Round 2
Reviewer 1 Report
All major flaws have been fixed. Authors need to pay attention to the formatting requirements and English spell cheking .
Author Response
Response to Reviewer 1 Comments
Point 1: All major flaws have been fixed. Authors need to pay attention to the formatting requirements and English spell cheking.
Response 1: Thank you so much for your comments. We have revised this manuscript for some formatting and English spelling problems.
Reviewer 2 Report
All my queries have been addressed now.
Author Response
Response to Reviewer 2 Comments
Point 1: All my queries have been addressed now.
Response 1: Thank you so much for your comments.